# K63 Ubiquitination of P21 Can Facilitate Pellino-1 in the Context of Chronic Obstructive Pulmonary Disease and Lung Cellular Senescence

**DOI:** 10.3390/cells11193115

**Published:** 2022-10-03

**Authors:** Jia-Hui Ma, Yi-Ting Zhang, Lu-Ping Wang, Qing-Yu Sun, Hao Zhang, Jian-Jiang Li, Ning-Ning Han, Yao-Yao Zhu, Xiao-Yu Xie, Xia Li

**Affiliations:** 1Marine College, Shandong University, Weihai 264200, China; 2School of Life Science and Technology, China Pharmaceutical University, Nanjing 210009, China; 3College of Biomedical Engineering and Instrumentation Science, Zhejiang University, Hangzhou 310000, China; 4School of Pharmaceutical Sciences, Shandong University, Jinan 250012, China

**Keywords:** COPD, lung senescence, Pellino-1, P21, ubiquitin

## Abstract

Chronic obstructive pulmonary diseases (COPD) is a kind of age-related, airflow-obstruction disease mostly caused by cigarette smoke. However, the relationship between COPD and lung cellular senescence is still not fully understood. Here, we found silencing Pellino-1 could inhibit the protein level of P21. Then, through constructing cell lines expressed ubiquitin-HA, we found that the E3 ubiquitin ligase Pellino-1 could bind to senescence marker p21 and modify p21 by K63-site ubiquitination by co-IP assays. Furthermore, we found that p21-mediated lung cellular senescence could be inhibited by silencing Pellino-1 in a D-galactose senescence mice model. Moreover, by constructing a COPD mouse model with shPellino-1 adenovirus, we found that silencing Pellino-1 could inhibit COPD and inflammation via reduction of SASPs regulated by p21. Taken together, our study findings elucidated that silencing E3 ligase Pellino-1 exhibits therapeutic potential for treatment to attenuate the progression of lung cellular senescence and COPD.

## 1. Introduction

Chronic obstructive pulmonary diseases (COPD) is regarded as a kind of chronic disease characterized by airway obstruction, which is largely irreversible [1]. Despite progress in the treatment of COPD symptoms, few advances have been made to ameliorate disease progression or mortality. The lung-cancer-related mortality of patients having COPD for longer than 5 years is increased 191% over that of patients without COPD [2]. Furthermore, the morbidity of lung cancer in COPD patients is increased more than five times vs. that of the healthy group [3]. Bronchodilation is the main treatment method used on COPD patients at present. Clinically, tiotropium bromide, an anticholinergic bronchodilator, most effectively improved the early stage of COPD patients [4]. Those patients who could not benefit from bronchodilators were treated with roflumilast or macrolides, which might cause severe side effects. The lack of effective drugs for COPD is an urgent matter and still presents a challenge.

The mechanisms leading to COPD mainly include inflammation, the imbalance of proteases and anti-proteases, oxidative stress, and cigarette smoke (CS) [5,6,7]. Among these mechanisms, CS is known as the primary risk factor of COPD and leads to premature lung cellular senescence [8,9,10]. Long-term CS accelerates COPD and the process of lung cellular senescence, resulting in increased damage to lung bubbles, airflow limitation, and a significant decline in lung function [11]. CS exposure destroys the epithelial barrier, leading to exposure of subepithelial layers to inspired air, which contributes to the progression of COPD [12]. Damage to the epithelial barrier would further lead to abnormal epithelial-mesenchymal transition (EMT), which could also promote lung cancer progression in COPD patients [13].

COPD is considered to accelerate the senescence process in the lung, and several senescence mechanisms are also applicable to COPD patients [14]. The relationship between COPD and lung cellular senescence is widely known, as oxidative stress causes cell cycle arrest and DNA damage mostly through the PI3K-mTOR or P38-SIRT1-P53/P16 pathways [15,16,17]. Reports have shown that p21 can regulate lung cellular senescence and COPD by inhibiting the phosphorylation of Rb [18]. Another important bridge linking COPD and lung cellular senescence is senescence-associated secretory phenotypes (SASPs). Senescent cells still maintain their metabolic functions and secrete SASPs, which alert their environments and senescent neighboring cells [19,20]. The SASPs, including IL-6, IL-8, and TNF-α, in COPD patients were significantly higher than those in the healthy patients [21]. Such SASPs, including certain cytokines, chemokines, proteases, and growth factors, are overexpressed in CS-induced COPD tissues, which could be mediated by p21 [16,22]. However, the type of modification of p21 that accelerated the senescence process and enhanced secretion of SASPs was still unknown.

Pellino proteins, as a family of E3 ubiquitin ligases, include Pellino-1, 2, 3 and play critical roles in regulating toll-like receptors and inflammatory pathways [23,24,25]. A study reported that Pellino-1 regulated the responses of the airway to viral infection but did not explain why this phenomenon occurred [26]. There are also reports showing that Pellino-1 could mediate lung cancer, but they did not report if Pellino-1 could regulate the transformation of inflammatory carcinoma, which might contribute to the research on Pellino-1 in lung cellular senescence and COPD [27,28]. Pellino-1, as an immuno-regulator, could enhance the secretion of many SASPs [29]. Interestingly, a review raised an opinion that Pellino-1 might be a regulator of lung injury through immuno-signaling [30]. However, there is still no research combining Pellino-1 with lung cellular senescence or COPD. 

Due to the relationship between COPD and lung cellular senescence, we studied the interaction of Pellino-1 and p21 and how they influence COPD and lung cellular senescence. Here, we found that ubiquitination of p21 was a critical senescence marker of COPD caused by CS. This research provided a new mechanism to prevent COPD and lung cellular senescence.

## 2. Materials and Methods

### 2.1. Cell Lines

Human normal lung cell lines BEAS-2B were cultured in RPMI-1640 medium (Hyclone, Logan, UT, USA) supplemented with 10% fetal bovine serum (FBS, Gibco, Carlsbad, CA, USA) at 37 °C in a 5% CO_2_ incubator. Human kidney epithelial cell 293T was cultured in DMEM medium (Hyclone, Logan, UT, USA) supplemented with 10% fetal bovine serum (FBS, Gibco, Carlsbad, CA, USA) at 37 °C in a 5% CO_2_ incubator. All cell lines culture condition contained 2 mM glutamine, 100 units/mL penicillin, and 100 μg/mL streptomycin. The identity of the cell lines was verified by Short Tandem Repeat (STR) analysis compared with ATCC database. Mycoplasma test was performed per month using qPCR assay.

### 2.2. Mice Experiments

For studies of lung senescence and COPD, we used 6-week-old BALB/c mice (male) in the beginning of this study, and divided them into 6 groups, with 8 mice each group, randomly. In order to investigate the role of PELI1 in lung senescence, we divided mice into a younger group, D-Galactose group, and elder group. Mice in the D-Galactose group were treated with D-Galactose (100 mg/kg) per day i.h. and sacrificed at the end of 9 weeks. The younger and elder groups were treated with an equal volume of PBS. The younger group was sacrificed at the end of 9 weeks and the elder group was sacrificed at the end of 18 weeks. Each group was treated with sh-PELI1 or EGFP adenovirus at the beginning of the experiment and the end of 6 weeks, respectively, i.v. In order to avoid losing effects of adenovirus, the elder group was injected additionally at the end of 12 weeks. For each mouse, 1 × 10^9^ PFU adenovirus solved in saline was injected in the caudal vein. The effectiveness of adenovirus was verified by blood mRNA test of Pellino-1. 

In order to investigate the role of PELI1 in COPD, we divided mice into an air group and a cigarette-smoking (CS) group, with 8 mice each group, randomly. Mice in the CS group were treated with 5 burned cigarettes (Taishan Brand, Jinan, China) per day for 18 weeks in a plexiglass whole-body exposure chamber (50 cm × 50 cm × 70 cm), with each cigarette containing 11 mg tar, 13 mg carbon monoxide, and 0.9 mg nicotine. Mice in the air group were not treated with CS. Mice were injected with sh-PELI1 or EGFP adenovirus at the beginning of the experiment and the end of 9 weeks, respectively, (i.v.). Mice were sacrificed at the end of 18 weeks. Before sacrifice, mice were treated with 10 mg/kg budesonide, a bronchodilator, and anesthetized by urethane. Then, a soft airflow duct (diameter-1 mm) was carefully injected into a small cut on the trachea. The lung function of FRC, FEV1, and FVC were tested using an auto small animal lung function testing system (BUXCO Fine Pointe system, DSI, DE, USA). 

All animal studies were approved by the Laboratory Animal Ethical and Welfare Committee of Shandong University Cheeloo College of Medicine (Permit No. 18013) and all procedures were conducted in accordance with the guidelines of Shandong University Cheeloo College of Medicine. All animal studies were abided by double-blind rules.

### 2.3. MTT Assay

MTT assay was used to analyze cell viability. Cells were seeded in 96-well plates, and after incubation and treatment, 15 µL MTT (5 g/L, Sigma, NJ, USA) were added to wells for 4 h. Then, DMSO were added after removing liquid. The absorbance of 570 nm was measured with a microplate reader (Molecular Devices, CA, USA). IC50 values were calculated by Graphpad Prism 9.2 software.

### 2.4. Transfection of siRNA

PELI1 siRNA and siRNA control was transfected into cells with Lipofectamine^®^ 2000 (Hanbio Biotechnology Co., Ltd., Shanghai, China). Control siRNA and PELI1-siRNA were constructed by Gene Pharma Company, Shanghai, China. Cells were seeded in 6-well plates and cultured for 24 h after transfection to be measured.

### 2.5. RT-qPCR

Total RNA was extracted with Trizol (Invitrogen, MA, USA, Cat# 15596026), and cDNA was synthesized with a fast reverse transcription kit (Sparkjade, Qingdao, China, Cat# AG0304). RT-qPCR was performed with a SPARKscript II RT Master Mix (Sparkjade, Qingdao, China, Cat# AG0202). The ΔΔCT method was used to analyze the fold of mRNA expression of GAPDH. RT-qPCR primers were shown in Appendix A.

### 2.6. Western Blot

RIPA buffer-medium (Beyotime, Cat# P0013C) was used for cells lysis. Lung tissue of mice was treated with a tissue grinder and lysis by RIPA buffer-strong (Beyotime, Cat# P0012). Total protein was quantified using a BCA Protein Assay Kit (Beyotime, Cat# P0012). Proteins were separated by SDS–PAGE gel electrophoresis and transferred to a nitrocellulose membrane (Bio-Rad, #1704271). Primary and secondary antibodies were listed in Appendix A. Membrane was treated with Sparkjade ECL super kit (Sparkjade, Qingdao, China, Cat# ED0015). Blot membranes were captured with a Fluor Chem M system (Protein Simple, CA, USA). Blot images were analyzed using Image J Software.

### 2.7. Analysis of Bronchoalveolar Lavage Fluid (BALF)

The lungs were lavaged via a cannula inserted into the trachea. To gain the fluid with cells, 2 × 1 mL aliquots of saline were instilled and then centrifuged at 2000 rpm for 10 min at 4 °C. The supernatants were obtained for ELISA analysis. Then, resuspended cells were counted in a hemocytometer. Cells were coated on glass slides and stained with Wright–Giemsa (Solarbio, Beijing, China) solution. Cells were differentiated by morphological criteria.

### 2.8. Cell Cycle Distribution

Cells stained with PI (Beyotime, Cat# C1052) were analyzed with flow cytometry (BD C6 Accuri, NY, USA) to measure cell cycle arrest condition. Results were analyzed with winMDI software.

### 2.9. Immunofluorescence Staining Assay

BEAS-2B cells were seeded on 6-well plates. Cells were permeabilized in 1% Triton X-100 for 15 min, and then blocked in 5% BSA for 1 h, washed twice before incubated with anti-PELI1 and anti-P21 antibodies overnight at 4 °C. Then goat anti-rabbit secondary antibody was used for 1 h at 37 °C and then washed and stained by DAPI for 10 min. Immunofluorescence images were capture by ZEISS Axio Observer (ZEISS, Oberkochen, Germany). Images were analyzed and treated with ZEISS ZEN lite software.

### 2.10. Immunoprecipitation

Following the published research, an immunoprecipitation (co-IP) assay was performed [31,32]. In brief, K63 or K48 HA-Ubiquitin plasmids were transfected into BEAS-2B cells. When stably expressing HA-Ubiquitin, cells were washed by PBS three times and collected in 550 μL Co-IP lysis (50 mM Tris (pH 7.5), 150 mM NaCl, 5 mg/mL aprotinin, 1 mg/mL pepstatin, 1% NP-40, 1 mM EDTA and 0.25% deoxycholate). After lysed for 30 min at 4 °C, liquids were centrifuged for 30 min at 12,000 rpm. The concentration of the supernatant was measured by BCA assay kit and normalized. The resin was then incubated with Protein A/G Plus–Agarose (Santa Cruz) beads at 4 °C for 2 h with gentle rocking. Then the immunocomplexes were mixed with 5× loading buffer and boiled for 10 min at 98 °C. The precipitated proteins were subjected to SDS-PAGE gel and analyzed with corresponding antibodies.

### 2.11. Degradation of P21

To determine protein degradation, cells were incubated with the protein synthesis inhibitor cycloheximide (CHX, 10 μg/mL) for the indicated times and the expression of indicated proteins were evaluated by immunoblotting and quantitative analyses.

### 2.12. Preparation of Cigarette Smoking Extraction (CSE)

Preparation of CSE was reported in our previous study [33]. Ten burned cigarettes (Taishan Brand, Jinan, China) were collected in PBS by a vacuum pump. Then CSE was filtrated by 0.22 μm filter and the CSE was mixed with medium containing 10% FBS before use.

### 2.13. Immunohistochemistry (IHC) Staining

Lung tissues were fixed in 3.7% formaldehyde diluted by PBS for 24 h and sliced by a Leica CM1520 freezing microtome. IHC was performed on 5-μm tissue sections using the ABC Kit after adding antibodies of P21. Diaminobenzidine (DAB) was used to dye the sections. Bronchial wall thickening was quantified by Image-pro Plus software (Media Cybernetics, MD, USA).

### 2.14. Adenovirus and Plasmids Construction

Plasmid of K48 and K63 ubiquitin were purchased from Addgene [34]. Empty control EGFP, shPELI1, and PELI1 overexpression adenovirus were constructed by Shanghai Hanbio Company, Shanghai, China. All adenovirus with EGFP (1 × 10^10^ PFU/mL) were infected to 293 cells. After 48 h culturing, cells were harvested, split, and treated with ST buffer. Then adenovirus was purified and stored in −80 °C.

### 2.15. Quantification and Statistical Analysis

Data are presented as the mean ± standard error of the mean (SEM). A student’s t test (2-tailed) was used to compare differences between the two groups. A one-way ANOVA test was used to analyze differences among multiple groups. *p* < 0.05 was considered statistically significant. Analyses were performed using Graphpad Prism 9.2 software (Graphpad, CA, USA). All experiments had been repeated at least three times (details seen in figure legends).

## 3. Results

### 3.1. E3 Ligase Pellino-1 Ubiquitylates P21 at the K63 Site

Pellino-1 (PELI1), an E3 ubiquitin ligase, had shown the ability to adjust immunosignals and metabolism [35,36,37]. In order to determine if PELI1 could regulate signals in the COPD process, we first screened a series of genes associated with COPD and found a positive correlation between Pellino-1 and p21 expression in the GTEx-lung database and TCGA para-carcinoma lung tissue database using GEPIA online tools (Figure 1A and Appendix A). Using pulmondb online analysis tools and GSE1650 datasets, we analyzed the relationship between PELI1 and p21 in COPD lung tissues (Figure 1B). The column similarity matrix showed a huge discrepancy in PELI1 and p21 between COPD patients and healthy people, while patients in same state of health showed obvious similarities. These bioinformatical analyses implied that PELI1 had a strong relationship with p21. Then, we investigated whether Pellino-1 participated in a protein–protein interaction with p21. The result of a co-IP assay showed that Pellino-1 could directly bind to p21 in both homo normal lung epithelial cell lines (BEAS-2B) and homo normal kidney epithelial cell lines (293T) (Figure 1C). Then, we used two kinds of siRNA to knockdown Pellino-1. In both BEAS-2B and 293T cell lines, knockdown Pellino-1 led to a decrement of p21 and an increment of p-Rb and Cyclin E (Figure 1D). Notably, except p21, the protein expression level of typical senescence marker P16 and P53 did not change significantly. We also found that the half-life of p21 was shortened by silencing Pellino-1 by treatment with 10 μg/mL of the protein synthesis inhibitor CHX (Figure 1F). However, we found that mRNA levels of p21 did not decrease after Pellino-1 knockdown in both BEAS-2B and 293T cells, which implied that pellino-1 may regulate p21 by degradation (Figure 1E). Moreover, we found after treatment with 5 μg/mL MG132 (an inhibitor of the 26S proteasome) and silencing of PELI1 that the protein level of p21 did not significantly change while that of p-Rb did increase, which enhanced the hypothesis that pellino-1 may regulate p21 by degradation (Figure 1G). Thus, to explain how Pellino-1 down-regulated the protein level of p21, we performed a series of assays to find the specific combination mode between them. SKP2, known as an E3 ligase bound to p21, was reported to regulate the p21 degradation via ubiquitin at the K48-Ub site [38,39]. After Pellino-1 silencing and MG132 treatment, we found that the binding of p21 and SKP2 was increased (Figure 1H). Furthermore, we established two kinds of BEAS-2B cells which stably expressed HA-targeted ubiquitin at K63 and K48 sites. We used HA tags antibody to perform the co-IP assay against P21 and Ub-HA proteins. In K63-Ub-type cells, treated with MG132, the results of co-IP assay showed that silencing Pellino-1 led to a decrease in HA-targeted K63-ubiquitin complexed with p21 (Figure 1I). While in K48-Ub-type cells, silencing Pellino-1 led to an increase in HA-targeted K63-ubiquitin complexed with p21 (Figure 1J). In this part, we clarified that the E3 ligase Pellino-1 could bind to p21 and link ubiquitin at the K63 site, which decreased SKP2-mediated K48-Ub-linked p21 and avoided the degradation of p21.

### 3.2. Silencing Pellino-1 Inhibits Senescence by P21-Mediated Cell-Cycle Arrest

A mass of research had reported that D-Galactose was widely used for constructing senescence models, which causes oxidative stressed senescence both in vitro and in vivo [40,41,42,43]. After treatment with five percent cigarette smoke extract (CSE) and D-Galactose (20 μg/mL), the cell viability of BEAS-2B was significantly decreased and the effect reversed by silencing Pellino-1 (Figure 2A). Additionally, we added MG132 to examine the influence of p21 degradation. MG132 treatment did not significantly influence cell viability. Silencing Pellino-1 decreased the overexpression of p21 protein caused by D-Galactose, which led to an increase in p-Rb and Cyclin E (Figure 2B). Silencing PELI1 did not change the p21 level but did alter p-Rb and Cyclin E levels in the presence of MG132, indicating that silencing PELI1 regulated p-Rb and Cyclin E via degradation of p21. The results of immunofluorescence staining showed the same tendency in the p21 protein level (Figure 2D). Moreover, CSE treatment showed the same tendency. Cyclin E and p-Rb were key regulators that kept the cell cycle stepping into S phase. Based on the change in p-Rb and Cyclin E, we used flow cytometry to investigate the cell cycle state. Treatment with D-Galactose and CSE arrested cells stepping into S phase and increased the ratio of G1 phase cells (Figure 2C). Then, we found that silencing Pellino-1 could inhibit G1 phase arrest and rebalance the cell cycle state. Moreover, using a β-Galactosidase staining assay, we found that silencing Pellino-1 could inhibit the senescence caused by D-Galactose, CSE, and MG132 (Figure 2D). With MG132 treatment, silencing Pellino-1 did not change the protein level of p21 but inhibited G1 phase arrest and cellular senescence, which indicated that Pellino-1 mediated cellular senescence through p21 degradation and its downstream effects on p-Rb and Cyclin E.

### 3.3. Silencing Pellino-1 Inhibits Senescence by Decreasing the P21 Level In Vivo

We then used adenovirus to infect shPELI1 plasmid and compared D-Galactose-mediated senescence in mice with natural senescence in younger and older mice to investigate the influence of Pellino-1 and senescence in vivo (Figure 3A). In the D-Galactose group and 18-week-old group, the movement and food intake of mice was obviously reduced, while the younger group and shPELI1 group did not show the same phenomenon. Comparing older and younger mice, the protein expression level of p21 was significantly elevated in older mice and could be reduced by shPELI1 (Figure 3B). The D-Galactose group showed a similar tendency. We also detected the expression of p-Rb and Cyclin E, which implied that silencing PELI1 would decrease the expression of p21 and lead to the recovery of the cell cycle regulators p-Rb and Cyclin E to inhibit the senescence process. We also found by immunohistochemistry assay that the p21 protein was overexpressed in the bronchia of the D-Galactose and older group (Figure 3D). Silencing PELI1 could reduce the expression of p21 and caused thinner bronchial walls. Then, using the HE staining assay, we found significant inflammatory infiltration in D-galactose-treated mice (Figure 3C). The older group showed less inflammatory infiltration but still more than younger group, which could be relieved by silencing PELI1. To further investigate why silencing PELI1 could relieve the inflammatory infiltration in senescent mice, we determined the level of superoxide dismutase (SOD) and malondialdehyde (MDA) in lung tissue (Figure 3E,F). The results showed that silencing PELI1 could relieve the reduction in SOD and overexpression of MDA caused by senescence. SASPs not only regulate the senescence process but also play a critical part in inflammation. Thus, we examined the mRNA expression of a series of SASPs and found that interleukin (IL)-6, IL-1α, matrix metalloproteinase (MMP)-9, MMP-12, tumor necrosis factor (TNF)-α, and chemokine CCL2 were significantly elevated in senescent mice and reduced by silencing PELI1 (Figure 3G). These results explained that PELI1 regulated lung cellular senescence through the oxidative stress signaling cycle of SASPs-p21-SASPs.

### 3.4. Silencing Pellino-1 Inhibits COPD and SASPs In Vivo

In a COPD mouse model, we also used adenovirus to infect shPELI1 plasmid to investigate the influence of Pellino-1 on COPD caused by cigarette smoke (Figure 4A). Functional residual capacity (FRC), forced expiratory volume in one second (FEV1), and forced vital capacity (FVC) were important parameters to evaluate patients’ lung function. COPD patients had a higher FRC and a lower ratio of FEV1/FVC. Before sacrificing the mice, we firstly gave 10 mg/kg budesonide, a bronchodilator, and investigated the difference in FRC and FEV1/FVC between CS and non-CS mice (Figure 4B). The results showed that silencing PELI1 could restore FRC and FEV1/FVC to a relatively normal level. Patients with COPD also showed symptoms of lung inflammation, which caused changes in the number of white cells. Then, total white cell and differential cell counts in bronchoalveolar lavage fluid (BALF) were performed (Figure 4C). We found that total white cells, macrophages, and neutrophils were significantly elevated in BALF of CS mice and could be reduced by silencing PELI1. Then, we used ELISA assay to check IL-6 and TNF-α levels in BALF and found that high levels of IL-6 and TNF-α caused by CS could also be reduced by silencing PELI1 (Figure 4D). Then, we found that the protein level of p21 in lung tissue was significantly higher in CS mice and could be reduced by silencing PELI1 (Figure 4F). Meanwhile, we determined the mRNA expression of Cyclin E (Figure 4E). Furthermore, using immunohistochemistry staining and HE staining, we found that silencing PELI1 could reduce inflammatory infiltration and the p21 expression level in bronchia and thinned bronchial walls caused by CS (Figure 4G,H). We also checked the mRNA levels of SASPs and MDA and SOD levels and found a similar tendency (Figure 5). Such results of senescence-related protein expression levels and pathological features in senescence and CS mice after silencing of PELI1 implied that PELI1 regulated COPD through the senescence signaling pathways. 

## 4. Discussion

Despite the infinite efforts of researchers and medical staff, COPD is still an incurable disease. Clinical medications for COPD are separated into three major classes: bronchodilators, anti-inflammatory agents, and antioxidants [44,45]. However, these treatments are only relievers, and relapse occurs easily. To date, the most effective COPD drugs are the inhaled dosage forms, which means that finding effective oral or i.v. drugs for COPD remains a challenge. One of the greatest challenges is the irreversibility of COPD due to its associated pathological changes and its enhancement effects on premature lung cellular senescence [44,46,47]. Cigarette smoking, the primary risk factor in COPD, could also lead directly to lung cellular senescence, which influences metabolism, oxidant stress, and endocrine signaling [48]. Meanwhile, COPD caused by cigarette smoke leads to significant overexpression of senescence markers, including p21, a CDK inhibitor. Regarding lung cellular senescence, p21 plays a critical part in regulating oxidative stress, the cell cycle, inflammation, and signal transduction [49,50,51]. Several E3 ligases can regulate the ubiquitination of p21, including SKP2, FBXO22, TRIM27, etc. [52,53]. However, none of these E3 ligases ubiquitinate p21 at the K63 site. Thus, in our research, Pellino-1 was unique for its ability to regulate p21. Due to a lack of animal experiments on COPD, which number far fewer than anti-cancer experiments, any progress on the mechanism of COPD is extremely valuable. Finding the combination of Pellino-1 and p21 might give us a new direction to design anti-senescence and anti-COPD medicine via suppression of the function of Pellino-1. 

Moreover, reports show that the irreversibility of COPD is closely associated with EMT leading to abnormal MMPs, growth factors, airway destruction, and remodeling [54,55]. These results were in close agreement with those of our former study that Pellino-1 could combine and regulate the EMT transcription factors snail/slug. That might, from another perspective, explain why pellino-1 could regulate COPD and lung cellular senescence.

In this research, we found the expression of Pellino-1 and P21 had a positive correlation. In order to investigate whether Pellino-1 could regulate the ubiquitin of P21, we first performed the co-IP assay and then found them to have a direct interaction. Then, we constructed two cell lines stably expressing Ub-HA at the K48 and K63 site, respectively. Using co-IP assay with HA-tags antibody, we found that the K63-ubiquitin of P21 was significantly decreased by siPellino-1. Meanwhile, the K48-ubiquitin of P21 was increased, which indicated that Pellino-1 regulated P21 through the K63 site ubiquitin modification of P21. Furthermore, we performed β-galactosidase staining assay and found siPellino-1 could inhibit the senescence of lung cells. There had already been experiments proving that P21 could promote the lung senescence and might influence the COPD progression. Thus, we constructed senescence-model in vivo, where we found siPellino-1 could efficiently inhibit the lung aging procession via the P21-related aging pathway. When COPD occurred, it always led to the lung senescence. On the other hand, a large proportion of old patients showed COPD symptoms. Based on these, we constructed a COPD model in vivo and found siPellino-1 could suppress the COPD progression. In both the COPD and senescence model, we found siPellino-1 could inhibit the mRNA expression of SASPs.

In summary, long-term cigarette smoke caused COPD and lung inflammation, which promoted the release of SASPs. Overexpressed SASPs led to the increment of P16, P53, and p21 protein levels. In the non-senescent condition, p21 was ubiquitinated by SKP2 and degraded by the 26S proteasome. In the senescent condition, p21 was ubiquitinated by Pellino-1 at the K63 site and inhibited the phosphorylation of Rb. Then, unphosphorylated Rb combined with E2F to inhibit the expression of Cyclin E, which led to G1 cell cycle arrest and accelerated cellular senescence. This signaling cycle explains how Pellino-1 regulates lung cellular senescence to a certain extent. The results of the present study provide the first evidence that the E3 ligase Pellino-1 could combine with p21 and regulate its K63-site ubiquitination.

## Figures and Tables

**Figure 1 cells-11-03115-f001:**
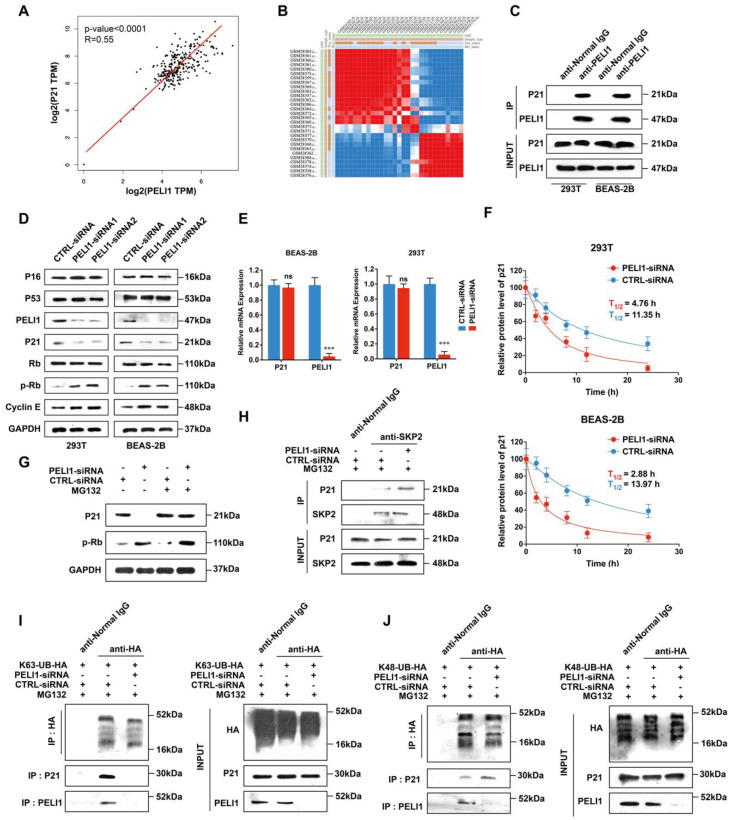
Pellino-1 regulates the degradation of p21 via K63-site ubiquitination. (**A**) The relationship between PELI1 and p21 was significant in the GTEx-lung database (Data represents R for relative coefficient and P for significance). (**B**) The column similarity matrix of PELI1 and p21 in COPD study GSE1650 showed a positive relationship in both COPD and healthy patients (Matrix values: pure blue for −1, pure red for 1). (**C**) Co-IP of PELI1 and p21 showed an interaction between PELI1 and p21 in both 293T and BEAS-2B cells (n = 3). (**D**) In 293T and BEAS-2B cells, two kinds of siRNAs of PELI1 both led to an increase in phosphorylated Rb and Cyclin E1 and a decrease in p21. P16, P53 and unphosphorylated Rb did not change significantly (n = 3). (**E**) The mRNA expression changes in p21 after siPELI1 did not change significantly. Data represent mean ± SEM; *** *p* < 0.001, ns for not significant (n = 3). (**F**) Quantitative analyses of p21 degradation in BEAS-2B and 293T cells after silencing of PELI1 following treatment with CHX (10 μg/mL). The half-lives (T_1/2_) of p21 degradation were calculated by single-phase decay analysis (n = 3). (**G**) Treatment with MG132 (5 μg/mL) and silencing of PELI1 led to an increase in p-Rb and did not change p21. (**H**) Co-IP assay showed that silencing PELI1 led to the enhanced combination of p21 and SKP2 (n = 3). (**I**,**J**) HA-tagged K48 and K63 ubiquitination sites were constructed. Silencing PELI1 led to a decrease in K63-Ub and an increase in K48-Ub combined with p21 (n = 3). Appendix A is related to Figure 1.

**Figure 2 cells-11-03115-f002:**
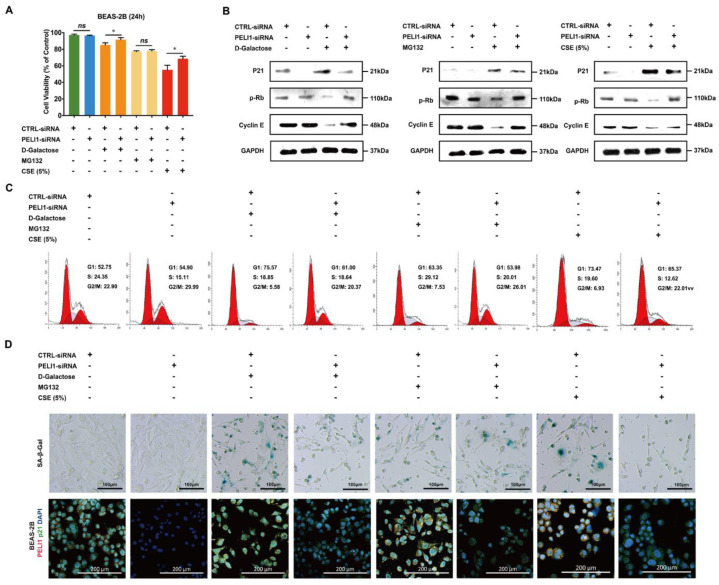
Silencing Pellino-1 inhibits G1\S arrest and cellular senescence caused by CSE and D-Galactose. (**A**) Treatment with CSE (5%) or D-Galactose (20 μg/mL) led to a decrease in the cell viability of BEAS-2B cells, which could be relieved by silencing PELI1. Data represent mean ± SEM; * *p* < 0.05, ns for not significant (n = 4). (**B**) CSE or D-Galactose treatment led to an increase in p21 and a decrease in p-Rb and Cyclin E, which could be relieved by silencing PELI1. Silencing PELI1 did not change the p21 level but did alter p-Rb and Cyclin E levels in cells treated with MG132 (n = 3). (**C**) The cell cycle of BEAS-2B showed G1 phase arrest following treatment with CSE and D-Galactose. Silencing PELI1 could help cells step into S phase (n = 3). (**D**) Up: SA-β-galactose staining showed that CSE, MG132, or D-Galactose treatment led to the cellular senescence of BEAS-2B. Such senescence could be inhibited by silencing PELI1 (n = 4). Down: Immunofluorescence staining assay showed that silencing PELI1 reduced the overexpressed p21 level caused by CSE or D-Galactose treatment. MG132 treatment did not produce the same effect.

**Figure 3 cells-11-03115-f003:**
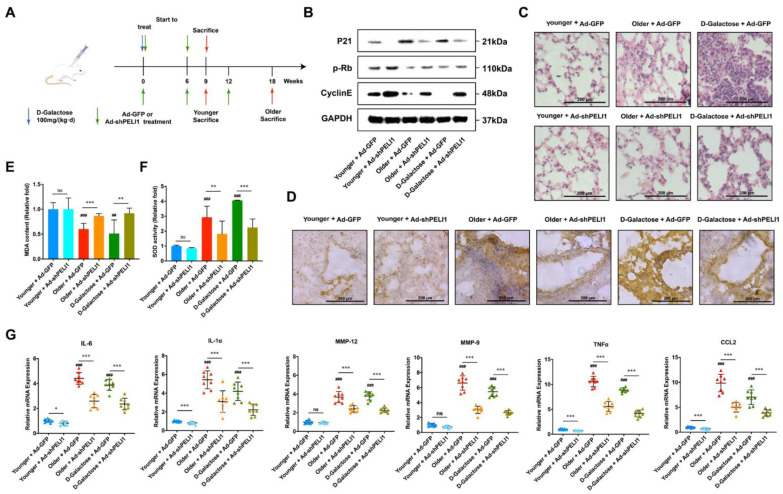
Silencing Pellino-1 inhibits lung cellular senescence via reduction of p21 and SASPs in vivo. (**A**) The strategy for studying the anti-senescence effects of silencing PELI1 in vivo. (**B**) Senescent mice showed an increase in p21, P16, and P53 and a decrease in p-Rb and Cyclin E, which could be reversed by silencing PELI1. (**C**) HE staining assay showed that silencing PELI1 reduced inflammatory infiltration in the pulmonary alveoli. (**D**) Immunohistochemistry showed that the bronchial wall thickening caused by senescence was reversed by silencing PELI1. Protein expression of p21 was also decreased by silencing PELI1. (**E**) MDA content of lung tissues. Silencing PELI1 relieved the reduction in MDA caused by senescence. (**F**) SOD activity of lung tissues. Silencing PELI1 relieved the increase in SOD activity caused by senescence. (**G**) The mRNA expression level of SASPs in lung tissue. Silencing PELI1 relieved the increase in SASPs caused by senescence. Data represents for E and F mean ± SEM; ^##^
*p* < 0.01, ^###^
*p* < 0.001 vs. (younger + ad-GFP group), * *p* < 0.05, ** *p* < 0.01, *** *p* < 0.001, ns for not significant, n = 8 mice.

**Figure 4 cells-11-03115-f004:**
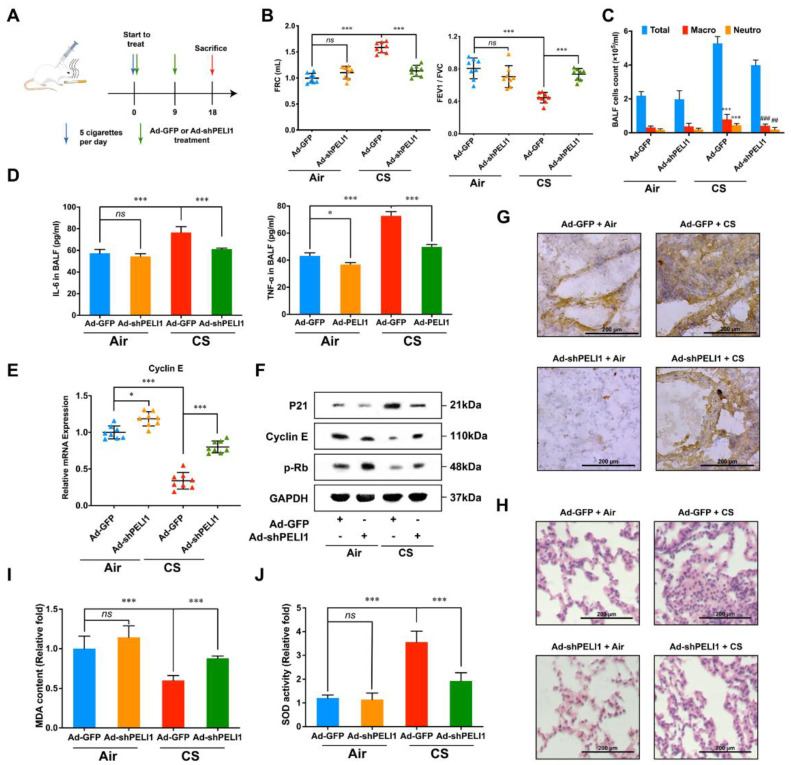
Silencing Pellino-1 inhibited COPD in anti-senescence and anti-inflammation pathways in vivo. (**A**) The strategy for studying the anti-COPD effects of silencing PELI1 in vivo. (**B**) FRC and FEV1/FVC following administration of 10 mg/kg budesonide 1 h before sacrifice. Silencing PELI1 decreased FRC and increased FEV1/FVC caused by CS significantly. Data represent mean ± SEM; *** *p* < 0.001, ns for not significant. (**C**) White cell counts in BALF showed that silencing PELI1 could decrease the high white cell counts caused by CS. Data represent mean ± SEM; ^##^
*p* < 0.01, ^###^
*p* < 0.001 vs. (CS + ad-GFP group), *** *p* < 0.001 vs. (Air + ad-GFP group). (**D**) ELISA to analyze the protein level of IL-6 and TNF-α in BALF. Silencing PELI1 significantly reduced the high protein levels caused by senescence. Data represent mean ± SEM; * *p* < 0.05, *** *p* < 0.001, ns for not significance. (**E**) Relative mRNA expression of Cyclin E was relieved by silencing PELI1. Data represent mean ± SEM; * *p* < 0.05, *** *p* < 0.001, ns for not significant. (**F**) Senescence-related protein levels were altered by silencing of PELI1. (**G**) Immunohistochemistry showed that the bronchial wall thickening caused by CS was reversed by silencing of PELI1. Protein expression of p21 was also decreased by silencing PELI1. (**H**) HE staining showed that inflammatory infiltration caused by CS in pulmonary alveoli was decreased by silencing of PELI1. (**I**,**J**) The change in MDA and SOD parameters caused by CS were relieved by silencing PELI1. Data represent mean ± SEM; *** *p* < 0.001, ns for not significant, n = 8 mice.

**Figure 5 cells-11-03115-f005:**
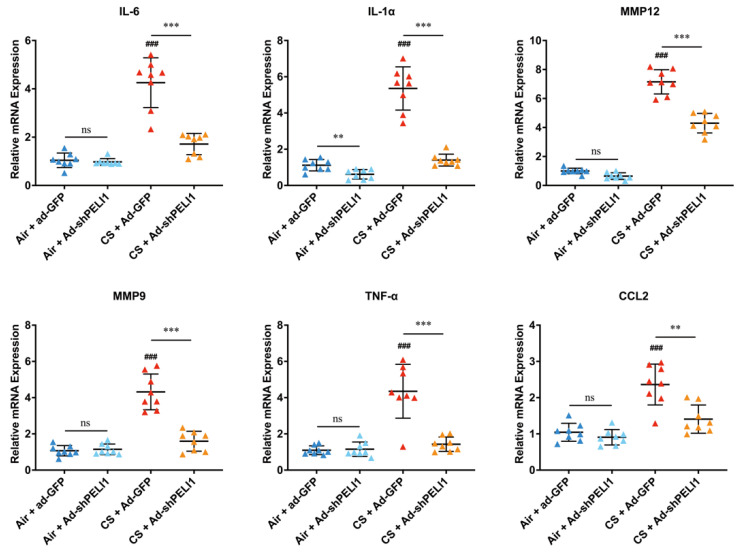
Silencing PELI1 decreased the high mRNA expression of SASPs caused by CS. ASPs including IL-6, IL-1α, MMP-9, MMP-12, TNFα, and CCL2 were significantly increased in the CS group, which could be decreased by silencing PELI1. Data represent mean ± SEM; ^###^
*p* < 0.001 vs. (Air + Ad-GFP group), ** *p* < 0.01, *** *p* < 0.001 vs. (CS + Ad-GFP group), ns for not significant, n = 8 mice.

## Data Availability

The data presented in this study are available on request from the corresponding author.

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
