# Peer review of "K63 Ubiquitination of P21 Can Facilitate Pellino-1 in the Context of Chronic Obstructive Pulmonary Disease and Lung Cellular Senescence"

_cells, 2022, doi:10.3390/cells11193115_

Round 1
Reviewer 1 Report
In this manuscript, the authors investigated the role of Pellino-1 in chronic obstructive pulmonary diseases (COPD) and lung cellular senescence. They found that Pellino-1(PELI1) can compete with SKP2 to ubiquitinate P21, thus preventing its proteasome degradation. They further showed that knockdown of PELI1 can inhibit G1/S arrest, cellular senescence, COPD, and the mRNA expression of SASPs. Overall, the physiology data presented in the manuscript is largely supporting the statements. However, there are still several points, especially the biochemistry part, need to be addressed before it can be published. For improvements of this manuscript, I have the following suggestions/comments:
Major points:
1. The biggest problem of the current manuscript is missing the direct genetic evidence that the function of Pellino-1 in lung cellular senescence and the related pathway is through its regulation on P21. In this regard, the authors need to manipulate the P21 protein level by either overexpressing P21 or knocking down/out SKP2 in siR-Pellino-1 background to see if it can rescue the phenotypes.
2. Fig.1, both I and J, I cannot tell very oblivious different ubiquitination levels between control and siR-Pellino-1, especially in the K48 blot. The authors mentioned in the legend they repeated these at least three times, I don’t understand why the representative images are still so low quality.
3. Fig.1I. Given the fact that siR-Pellino-1 knockdown was so efficient, how can it only cause a small reduction of K63 ubiquitination? This directly argues against the role of Pellino-1 in P21 ubiquitination.
4. Same thing applies to Fig.1J, with almost 2-3 fold more enrichment of P21-SKP2 interaction after Pellino-1 knockdown, how come it nearly has no effect on K48 ubiquitination?
5. In this ub-blot system, the smallest band detected should be 20+8 KD, why the authors detected 20KD or even smaller bands?
Reviewer 2 Report
Chronic obstructive pulmonary disease (COPD) is regarded as a chronic disease characterized by airway obstruction, which is largely irreversible mostly caused by cigarette smoke. Despite progress in the treatment of COPD symptoms, few advances have been made to ameliorate disease progression. However, the relationship between COPD and lung cellular senescence is still not fully understood. Authors found silencing Pellino-1 could inhibit the protein level of P21. Further they expressed ubiquitin-HA and observed the E3 ubiquitin ligase Pellino-1 could bind to senescence marker p21 and modify p21 by K63-site ubiquitination by co-IP assays. Interestingly silencing pellino-1 could inhibit COPD and inflammation via reduction of SASPs regulated by p21. They, found that ubiquitination of p21 was a critical senescence marker of COPD caused by CS. This research is novel and provided new mechanism to prevent COPD and lung cellular senescence.
Major comments –
1) Did the authors check the protein or mRNA expression of ATM (ataxia-telangiectasia mutated)? As, the p53 activation is dependent on the protein kinase ATM.
2) There are numerous grammar errors, incorrect tense uses, and typographical errors.
Round 2
Reviewer 1 Report
In this revised manuscript, the authors have addressed some of the concerns raised by this reviewer. However, there are still problems, including some fundamental ones that need to be addressed before it can be published. Specifically,
1. Following point 1 in my last round of comments. The key conclusion from one paper should be drawn from its data, not from the knowledge of other studies. As I mentioned in the last round, Pellino-1 regulates the protein level of p21, Pellino-1 regulates lung cellular senescence, and p21 is involved in lung cellular senescence are three independent results, and all of them were obtained from this manuscript, the authors should not mix them to make one conclusion in the title. I understand that due to the time limit, the authors may not have time to perform the experiments as suggested. But at least they should revise the title properly and add enough comments in the discussion section.
2. Following point 5 in my last round of comments. The reply from the authors make me even more confused. Do they really understand how ub-blot works? If they did the IP using HA beads, then probe for the P21, then P21 should only be detected starting around 30KD as the mono-ubiquitinated. On top of that, they should also observe higher molecular bands indicating poly-ubiquitination form in the p21 blot. However, they only showed a single band in the p21 blot labeled with 21KD. Without conjugated Ub-HA, how does p21 pulled down and detected? I suggest the authors:
1) Include detailed information about the ub-blot in the method section to make it clear how they did the experiment.
2) Check their design and results carefully to make the right interpretation. I suggest reading the following two papers (if necessary, also cite and use them as references to optimize their system) : PMID: 29414787, 34297722. One is using ubiquitin antibodies and the other is using HA-ub as the authors did in this manuscript.
Round 3
Reviewer 1 Report
In this revision the authors have addressed most of the concerns, now it is ready for publication.